# Intra- and Inter-Observer Reliability of Qualitative Behaviour Assessments of Housed Sheep in Norway

**DOI:** 10.3390/ani9080569

**Published:** 2019-08-17

**Authors:** Sofia Diaz-Lundahl, Selina Hellestveit, Solveig Marie Stubsjøen, Clare J. Phythian, Randi Oppermann Moe, Karianne Muri

**Affiliations:** 1Department of Production Animal Clinical Sciences, Faculty of Veterinary Medicine, Norwegian University of Life Sciences, P.O. Box 369 Sentrum, 0102 Oslo, Norway; 2Department of Basic Sciences and Aquatic Medicine, Faculty of Veterinary Medicine, Norwegian University of Life Sciences, P.O. Box 369 Sentrum, 0102 Oslo, Norway; 3Department of Animal Health and Food Safety, Norwegian Veterinary Institute, P.O. Box 750 Sentrum, 0106 Oslo, Norway; 4Section for Small Ruminant Research, Department of Production Animal Clinical Sciences, Faculty of Veterinary Medicine, Norwegian University of Life Sciences, 4325 Sandnes, Norway

**Keywords:** sheep, qualitative behaviour assessment (QBA), welfare assessment protocol, observer reliability, housing, animal welfare

## Abstract

**Simple Summary:**

Qualitative behaviour assessment (QBA) is a whole-animal approach to measuring animal welfare, based on observing the animal’s body language and behaviour. The method is used in different animal welfare protocols such as the Welfare Quality^®^ (WQ^®^) protocols developed for poultry, cattle and swine and the AWIN protocols for sheep and goats. In Norway, farmed sheep are typically housed during the winter period for approximately six months and this presents specific risks for animal welfare, as well as specific opportunities for improvement. A welfare protocol for sheep managed under Norwegian housing conditions was developed for the Norwegian Sheep House (*FåreBygg*) project. In this study, we tested the reliability of QBA as developed for this protocol, when used by six trained observers with different professional background and experience, using video recordings. Intra-observer reliability was assessed by viewing the videos twice with a one-week interval between viewings. The statistical analyses revealed high agreement between all observers, and between scorings of the same observers at different time points. The results suggest that the tested protocol is reliable for assessing video recordings of sheep behaviour when applied by trained observers, regardless of their occupation with differing experiences of sheep health, welfare and production.

**Abstract:**

This study tested the reliability of a Qualitative Behavioural Assessment (QBA) protocol developed for the Norwegian Sheep House (*FåreBygg*) project. The aim was to verify whether QBA scores were consistent between different observers, i.e., inter-observer reliability, and between scorings of the same observers on different time points, i.e., intra-observer reliability. Six trained observers, including two veterinary students, two animal welfare inspectors and two sheep farmers observed sheep in 16 videos, and independently scored 14 pre-defined behavioural descriptors on visual analogue scales (VAS). The procedure was repeated one week after the first scoring session. QBA scores were analysed using Principal Component Analysis. Inter- and intra-observer agreement was assessed using Kendall’s coefficient of concordance (*W*). Principal component 1 (PC 1) and 2 (PC 2) combined explained >60% of the total variation in the QBA scores in both scoring sessions. PC 1 (44.5% in sessions 1 and 2) ranged from the positive descriptors calm, content, relaxed and friendly to the negative descriptors uneasy, vigilant and fearful, and was therefore labelled mood. PC 2 (18% in session 1, 16.6% in session 2) ranged from bright to dejected and apathetic, and was therefore labelled arousal. Kendall’s coefficient of concordance of PC 1 for all observers was high in the two scoring sessions (*W* = 0.87 and 0.85, respectively), indicating good inter-observer reliability. For PC 2, the agreement for all observers was moderate in both video sessions (*W* = 0.45 and 0.65). The intra-observer agreement was very high for all observers for PC 1 (*W* > 0.9) except for one, where the agreement was considered to be high (*W* = 0.89). For PC 2, Kendall’s coefficient was very high for the veterinary students and interpreted as moderate for the two farmers and welfare inspectors. This study indicates that the QBA approach and the terms included in the *Fårebygg* protocol were reliable for assessing video recordings of sheep behaviour when applied by trained observers, regardless of whether they were a veterinary student, animal welfare inspector or sheep farmer. Further work is needed to examine the reliability of the QBA protocol when tested on-farms for sheep managed under Norwegian housing systems.

## 1. Introduction

In many developed countries, consumers of meat, milk, eggs and other products from animals are paying increasing attention to animal welfare, thus creating a demand for animal welfare-friendly products [1,2,3]. Consequently, there is an increasing need for the assessment. Standardized animal welfare assessment protocols incorporating valid, reliable and feasible welfare indicators provide a means of comparing the welfare of animals managed across different farms in a transparent, fair and consistent manner [4].

The more recently developed protocols typically consist of a mixture of animal-based and resource-based measurements [5]. Resource-based measurements focus on what is provided to the animal, including the environment, e.g., feed quality, spacing, hygiene and relative humidity. Animal-based indicators, on the other hand, reflect the response of an animal to a specific situation or environment [6,7]. Animal-based welfare outcomes provide a better insight into how different production and management systems influence and impact animal welfare compared to measures of resource and management inputs. Traditionally, measures of good animal health have been interpreted as a reflection of good animal welfare. For example, body condition score (BCS) has been identified as a valid, reliable and feasible animal-based indicator of sheep health and welfare [8,9,10]. However, a sheep in an ideal body condition is not necessarily experiencing a state of emotional well-being. Hence when measured alone, BCS and other measures of the animals’ physical condition do not provide sufficient information to draw conclusions about the actual welfare state of the animal. By comparison, qualitative behaviour assessment (QBA) is a whole-animal approach, based on observing the animal’s behaviour and body language [11]. One of the advantages of QBA is that the expression of positive emotions are taken into consideration, as opposed to only focusing on the presence or absence of signs of disease, pain or suffering [11,12].

QBA involves the use of behavioural descriptors, such as fearful, social, uneasy and content, to describe the welfare of an individual animal or a group of animals [13]. Originally, QBA was based on free choice profiling (FCP), whereby observers generated their own terms to describe the behaviour of animals, and then scored the intensity of each term along a visual analogue scale (VAS) [14,15]. This approach requires that the data are analysed using generalised Procrustes analysis (GPA). Other users of QBA have developed a fixed-list (FL) of predefined terms for the species of interest. Observers then score each of the terms on a visual analogue scale (VAS) (e.g., Welfare Quality Protocol for cattle). To date, principal components analysis (PCA) has been the most common approach to analyse data from fixed-list QBA, as it reveals the underlying structure of the data and reduces the number of variables to a few main dimensions, which can be interpreted in terms of animal welfare [12,16,17,18].

In recent years, QBA has been included in several welfare protocols such as the Welfare Quality^®^ (WQ^®^) protocols developed for poultry, cattle and swine [19] and in the AWIN protocols for sheep [20] and goats [21]. Various studies have investigated the validity of QBA, based on both FCP [22,23] and the FL approach [24]. For example, Stockman and colleagues [22] found that cattle that were assessed as agitated, restless and stressed during transport also had increased heart rate, core body temperature and plasma glucose compared to before transport. Whilst in a study of sheep, Wickham and colleagues [23] found that sheep described as more active and alert compared to the control when subjected to different stressors, showed increases in core body temperature, leptin concentration and haematocrit. The described changes in physiological parameters in both studies are consistent with a stress response, and supports the validity of the method in detecting complexities of behavioural expressions of animals.

Previous studies have found varying levels of reliability when QBA was performed on cattle using FCP [24,25], on pigs using FCP [26], and FL [12]. Methods of assessment used to test observer reliability vary from video-scoring [12] to direct observations of animals on-farm [24,25] or in experimental conditions [26]

Several reliability studies have been conducted on QBA for small ruminants [16,17,27]. In 2013, the first reliability study of QBA on sheep, based on observation of video clips found high inter-observer reliability [27].

QBA is based on the ability to interpret an animal’s body language, and has been suggested to be a supplementary welfare screening tool [15]. Studies on sheep [18] and goats [16], suggest that QBA is sensitive to detecting between-farm differences in animal health and welfare outcomes. To assess whether QBA is reliable and is externally valid for application by different stakeholders with different backgrounds in sheep health and welfare, it is important that the method is tested by the range of potential on-farm users, such as farmers, veterinarians and farm animal welfare inspectors. The method also needs to be tested under the range of management and farming types found in the region of interest.

In Norway, farmed sheep are typically housed for approximately six months during the cold season (October to April) and this presents opportunities for close inspection of sheep behaviour and stockperson attention. However, there may also be specific welfare risks associated with a long period of indoor housing. To date, the reliability of QBA when applied by a range of stakeholders for assessing sheep managed under Norwegian housing conditions has not been explored.

The background to this study was that QBA was one of a number of animal-based indicators that were included in an on-farm welfare assessment protocol used in the SheepHouse (*FåreBygg*) project. The project had a broader aim of examining the on-farm welfare of sheep managed in the different housing systems found in Norway using a combination of valid, reliable and feasible animal- resource- and management-based indicators (Norwegian Research Council Project nr. 225353/E40).

The objective of the present study was to examine the inter- and intra-observer reliability of the QBA protocol for housed sheep managed under Norwegian housing conditions, when a group of farmers, veterinary students and farm animal welfare inspectors applied the method to video recordings.

## 2. Materials and Methods

### 2.1. Experimental Design

#### 2.1.1. Behavioural Descriptors

The behavioural descriptors used in this study were modified from a previous QBA protocol for housed sheep, that included 12 descriptors created and used by Muri and Stubsjøen [17]. A focus group including five veterinarians with practical experience in sheep behaviour and welfare assessments, and two final year veterinary students discussed the descriptors in various steps; after watching both videos of sheep and sheep on-farm. This involved the inclusion and testing of new descriptors. Proposed changes were based on consensus about the definition, judgement and feasibility, as well as the term’s meaning for sheep welfare. All terms were tested on-farm followed by new discussions and revision of the list of terms. For each descriptor, the final protocol contained a written definition that the group had agreed upon.

Fourteen behavioural descriptors were finally included in this study (Table 1), of which half were negative terms, and the other half were positive terms. One behavioural descriptor, trustful, was excluded from the reliability study because there were no stockpeople present in most videos, making the assessment of trustfulness impossible. However, the term was kept in the *FåreBygg* protocol used on-farm [28]. For the remainder of this paper we only refer to the 13 descriptors included in the reliability study.

#### 2.1.2. Observers

Six observers with different backgrounds participated in the study. Two were final year veterinary students (authors S.D-L. and S.H.) of the Norwegian University of Life Sciences (NMBU), two were experienced full-time sheep farmers and two were experienced animal welfare inspectors and veterinarians from the Norwegian Food Safety Authority (NFSA). The students had received more training in using QBA than the other observers, as they had participated in group discussions and on-farm testing for development of the QBA protocol used in this study. The veterinarians from the NFSA were familiar with the method from a former study the previous year [17]. The farmers had not received any training in the use of QBA prior to the introductory presentation given at the first video scoring session.

#### 2.1.3. Videos

All, except for four, video recordings used in this study included recordings of adult sheep in the mid-east of Norway during the winter indoor-housing period. Videos of groups of sheep were recorded on 11 farms in connection with an earlier study [10] during a two-week period in April and May 2007. Average flock size on these farms was 111 (range 35–240), and the Norwegian White Sheep breed was kept on all but one farm that also had the old Norwegian breed *Spælsau*. The videos were selected by two of the scientists (S.M.S. and K.M.), and fourteen two-minute clips were chosen from this material, aiming to cover a variety of behavioural expressions observed in Norwegian sheep housing systems. To supplement the videos from Norwegian farms, four additional videos belonging to Professor Françoise Wemelsfelder from Scotland’s Rural College (SRUC) were used (videos number 2, 6, 14 and 16). The sheep on these videos were Scottish Blackface (clips 2, 6 and 16) or mule sheep (clip 16). In total, 16 videos were included in the reliability study. A brief description of each video is presented in Table 2.

#### 2.1.4. Study Procedure

The video sessions took place at NMBU Sandnes, section for small ruminant research, in February 2015. Before scoring started, an introductory presentation was provided by the test leader and the first authors. The behavioural terms and definitions (Table 1) were presented to the group of observers and discussed for 15 min to provide clarity. The written definitions were available to the observers throughout the period of the test. Each observer received 16 scoring sheets (one for each video) with a 125 mm visual analogue scale (VAS) from minimum to maximum below each descriptor. Minimum was defined as the level where an expression is not present among the observed animals at all, whereas maximum is the level at which the expression is dominant across the entire group being observed. The observers marked along the scale for each term, answering the question “How dominant is this behavioural term among the observed animals during the observational period?”

In 15 of the 16 videos, the observers were asked to observe all of the sheep that were visible during the video clip. In one video, they were asked to focus their observations on a group of three adult sheep (clip number 16). After each video, the observers had a few minutes to score each behavioural descriptor on the separate VAS provided for each clip. The same procedure was repeated one week later in order to assess intra-observer reliability, but the order of the video clips was changed using random number allocation.

### 2.2. Data Management and Statistical Analysis

VAS data were transferred to a spreadsheet (Microsoft Office Excel^®^ 2010) by recording the distance in millimetres from the minimum mark to the point where the scale was ticked.

Data were analysed in STATA SE/12.1 (StataCorp, College Station, TX, USA).

PCA based on a correlation matrix (no rotation) was conducted on the data from the two scoring sessions. The results from PCA are presented as component scores, which describe the total variation of data in the main dimensions. The first two principal components from the analyses (PC 1, PC 2) were retained for further analysis in both sessions, based on a combination of the elbow plot criterion and Kaiser’s criterion [29].

Subsequently, the component scores for the two principle components were used to assess the agreement between observers (inter-observer reliability) in the two scoring sessions, and for each observer on different days (intra-observer reliability). Agreement was assessed using Kendall’s coefficient of concordance (*W*). The results were interpreted according to Martin and Bateson [30], where *W* = 0.0–0.2 indicates a slight correlation, *W* = 0.2–0.4 a low correlation, *W* = 0.4–0.7 a moderate correlation, *W* = 0.7–0.9 a high correlation, and *W* = 0.9–1.0 indicates a very high correlation.

As different observers had different backgrounds and experience with the method, the inter-observer reliability was also investigated within the following pairs or groups: (1) veterinary students, (2) animal welfare inspectors from NFSA, (3) sheep farmers, (4) veterinary students and animal welfare inspectors from NFSA combined, and (5) all observers combined.

## 3. Results

Principal component 1 and 2 combined explained >60% of the total variation in the QBA scores in both scoring sessions (62.5% in session one and 61.1% in session two). In the first scoring session, PC 1 (eigenvalue 5.8) explained 44.5% of the variation while PC 2 (eigenvalue 2.3) explained 18.0% of the variation. In the second scoring session, the variation explained was 44.5% for PC 1 (eigenvalue 5.8) and 16.6% for PC 2 (eigenvalue 2.2).

The loadings of each behavioural descriptor along the PC 1 and PC 2 from the first and second scoring session are shown in Figure 1 and Figure 2, respectively. In the loading plot for all of the observers, PC 1 ranges from the positive descriptors calm, content, relaxed and friendly to the negative descriptors uneasy, vigilant and fearful, and was therefore labelled mood. PC 2 ranged from bright to dejected and apathetic (Figure 1), and was therefore labelled arousal.

### 3.1. Inter-Observer Reliability

Table 3 presents *W* values and p-values from the analyses of the inter-observer reliability within the different observer groups and overall. For PC 1, *W* values were >0.75 for all observer groups, in both video scoring sessions. For PC 2, the agreement ranged from *W* = 0.45 (all observers, session 1) to *W* = 0.91 (veterinary students, session 1).

### 3.2. Intra-Observer Reliability

Table 4 illustrates that for PC 1, the intra-observer agreement was very high for all observers (*W* > 0.9) except for one, where the agreement was high (*W* = 0.89). For PC 2, Kendall’s coefficient was very high (>0.9) for the vet students and interpreted as moderate for the two farmers and NFSA inspectors (ranging from 0.45 to 0.67) (Table 4).

## 4. Discussion

This study assessed both inter- and intra-observer reliability of a fixed list QBA protocol for sheep under Norwegian housing conditions, based on video recordings. It has been suggested earlier that sheep hide signs of distress and pain and that human observers might have difficulties interpreting their behavioural expressions [10], thus underlining the need for reliable behavioural methods for inclusion in animal welfare assessment protocols.

### 4.1. Dimensionality of Qualitative Behavioural Assessments

PC 1 ranged from the positive descriptors calm, content, relaxed and friendly to the negative descriptors uneasy, vigilant and fearful. This summarizes the moods expressed by the sheep in the video clips and is consistent with the general tendencies in comparable studies of sheep [17,27]. The anchoring points for PC 2 are also somewhat comparable to the anchoring points identified earlier by Phythian and co-workers [27] and Muri and Stubsjøen [17], describing the arousal (bright to dejected, and apathetic). QBA studies of other species (donkeys [31], goats [16] and cattle [24] using fixed-list approach also suggests that the two main components describes mood (PC 1) and arousal (PC 2).

### 4.2. Inter- and Intra-Observer Reliability

All pairs of (1) veterinary students, (2) animal welfare inspectors, (3) sheep farmers, and grouping of (4) veterinary students and animal welfare inspectors combined, and (5) all observers combined obtained acceptable inter-observer reliability for PC 1. The observer pairs 1) vet students and 3) farmers reached a very high level of agreement. Comparing all observers combined, showed a high level of correlation for PC 1. Previous reliability studies have shown excellent (*W* > 0.9) and high (*W* > 0.7) inter-observer reliability for sheep [27] and [17], respectivly. There is divergence in the findings for cattle ranging from high levels of between observer agreement [24] to poor [25]. In the present study, agreement between the two welfare inspectors decreased from very high in session 1 to moderate in session 2, while correlation remained very high in the two other groups in both sessions. If one or both of the welfare inspectors did not calibrate with the pre-agreed fixed definition, this could also have caused the disagreement.

The inter-observer reliability of PC 2 was interpreted as acceptable (*W* > 0.7) for the veterinary students and sheep farmers in both sessions, and for NFSA inspectors in the second session. For the all-observers group, the inter-observer reliability of PC 2 was interpreted as not acceptable (*W* < 0.7) in both sessions, suggesting that the second component may not be sufficiently reliable using the current protocol. PC 2 was interpreted as moderate (*W* = 0.69) in a study of sheep using a previous version of the same QBA-protocol [17]. This differs slightly from the earlier work of Phythian and colleagues [27], who found higher reliability for PC 2 between all assessor groups using pre-fixed terms.

The intra-observer correlation of PC 1 was >0.7 for all observers, thus considered as acceptable reliability. However, there was variation amongst the observers. Agreement for one of the animal welfare inspectors was lower than that of the other observers. The second dimension on PC 2 showed the same tendencies as in the inter-observer study, with some groups showing a non-acceptable level of reliability.

### 4.3. Observer Effects: Profession, Experience and Previous Training

The current study did not identify any significant differences in the reliability of QBA when applied by different groups of observers. The groups differed in their level of training or time elapsed since training, but they also represented different professions i.e., sheep farmers, veterinary students and qualified vets working as animal welfare inspectors, with a variety of different experiences and perspectives on sheep health, welfare and behaviour.

The intra-observer reliability was equally high for all individuals independent of category (“student, farmer, welfare inspector”), suggesting that QBA could be applied by these different stakeholders to score videos of sheep behaviour with a high level of individual consistency. In this study, all observers had professions or education related to animal health or welfare. However, Duijvesteijn and colleagues [12], found that even more diverse groups (pig farmers, animal scientists and urban citizens) applying QBA to assess video clips of pigs achieved equally high level of intra-observer reliability (correlations of 0.6–0.7, using a correlation circle), regardless of their prior experience with pigs.

The inter-observer reliability for PC 1 in our study was not only high within the observer groups, but also when all-observers results were combined, suggesting that the observers, independent of their professional background with farm animal health, welfare and production, had a similar way of scoring the videos. Our results are in agreement with the sheep QBA video study of Phythian and colleagues, who also found high levels of inter-observer reliability between the groups of veterinary students/veterinarians and farm assurance assessors [27].

However, groups with other professional backgrounds may put emphasis on different aspects of animal behaviour and welfare. Duijvesteijn and colleagues [12] found significant differences in the scorings of the first dimension between farmers, researchers and urban citizens, thus indicating that there was poor between-observer agreement between these stakeholders. In that study, the varied observer groups were thought to represent different views of animal welfare: farmers seemed to have a more positive interpretation of the pigs’ behaviour than the two other groups in general. The current study did not indicate the same tendencies on inter-observer reliability. However, due to logistical and time resources, only two observers were represented per professional group, which limited the statistical meaningfulness and generalisability of this result.

The level of QBA training did not seem to have an effect on the reliability of PC 1. All groups received an introductory presentation of QBA and the use of VAS scales. The veterinary students had received more training in QBA than the full-time sheep farmers, but the farmers had a much broader and longer practical experience with working with sheep managed in Norway. Others have found that experience with the species of interest and training of assessors has a considerable effect on reliability. Bokkers and colleagues [25] identified lower levels of reliability in the less experienced group performing QBA of dairy cattle, whereas Andreasen and colleagues [24] found that observers with only one day of training in the method had a very high inter-observer reliability when applying the same QBA protocol as Bokkers et al. Similarly, for sheep, Phythian and colleagues [27], found high inter-observer agreement for assessors with one day of training. Whilst the present study population was too small to separate the effect of professional background from the effect of QBA training, the results might suggest that QBA is intuitive for those experienced with sheep management as well as those less experienced observer groups that receive sufficient training in the specific species of interest.

There was more variation found in between-observer reliability for the second dimension (PC 2). Veterinary students reached a very high agreement for both intra- and inter-observer reliability, while the other observer groups provided moderate to high agreement for PC 2. Flemming and colleagues [14] found similar results for sheep, and suggested that sufficient observer calibration and training, including practice in the use of VAS scales prior to video scoring, was an important factor for achieving good intra-observer reliability.

### 4.4. Study Limitations

Assessment of intra-observer reliability may have been limited due to the relatively short interval (one week) between the two scoring sessions. It is possible that observers replicated the last scoring rather than assessing the videos intuitively. However, guidance on studies of diagnostic reliability [32] was followed and the order of the videos was changed in the second session to reduce this effect.

Whilst a longer interval between videos may be useful, increasing the time between the sessions would also increase the risk of observer drift, i.e., the observers altering their personal understandings of the definitions unconsciously [30].

Some scientists have considered QBA to being anthropomorphic and unscientific due to the apparent subjectivity in this approach [11]. QBA uses qualitative descriptors, but that does not necessarily mean that the method is more subjective than other methods based on observer judgement [33]. In QBA, there is a qualitative element not only in the interpretation of the results, but also when making the measurements. However, this is also true for other animal-based indicators that rely on subjective assessments such as scoring the severity of skin lesions. The expression animal welfare and our judgement of it, has its derivation in the human ability to perceive and interpret complex body language and behavioral signs [11]. Due to this, the integrative nature of QBA might be a good and even essential thing. If scientists only use quantitative measures when assessing animal welfare, they might risk overlooking important information [34], since some aspects of the concept are difficult to quantify.

Tuyttens and colleagues [35] identified that presenting observers with positive or negative information prior to QBA of cattle, pigs and laying hens resulted in significant expectation bias. From studies of sheep transported in Australia, Fleming and colleagues [14] concluded that QBA was influenced by observer bias, but the comparative ranking of animals (the pattern of interpretation), using multivariate techniques like PCA was not influenced. These results suggest that observer bias does not completely change the judgement of behaviour, and that it can be reduced by ensuring that as little additional information as possible is presented to observers prior to QBA of video recordings.

In observer ratings, expectation bias may occur for other reasons, such as a change of environmental setting. Wemelsfelder and colleagues [36] found that the observers’ interpretation of pig behaviour using FCP was slightly affected by digitally altering the background environment. However, the pattern of interpretation of pig behaviour was stable. Hence, it was concluded that the contextual sensitivity of the method, is unlikely to weaken the reliability of QBA in general. Expectation bias was probably reduced in this study, because the observers were not presented to other conditions on the farm such as the condition of the barn in general and surrounding environment, or the attitude of the farmer.

The current study suggests that the QBA approach employed within the *FåreBygg* protocol is reliable for video scoring of sheep under Norwegian sheep farming conditions. Previous QBA studies of sheep have reported that higher levels of observer reliability were achieved through video recording compared to on-farm QBA. Video scoring presents controlled conditions for reliability testing but cannot fully represent an on-farm situation, and creates a challenge in comparing reliability of the two approaches. However, this issue that remains for all reliability studies based on video scoring. The next step would be to test the reliability of the same QBA protocol on-farm. This might require some alterations of the method due to several reasons. For example, in our study the videos lasted for about two minutes. In the on-farm protocol of the project to which this study belongs (*FåreBygg*), QBA was assessed for 20 min. It is possible that a longer period of observation is needed during on-farm assessments in order to capture the variety of behavioural expressions in large groups of animals, sometimes divided in different pens, rooms or buildings. This also requires that the observers changes their physical position a few times, and the presence of the observer might therefore be a disturbing factor for the animals. The videos were chosen to cover a range of different dominant aspects of sheep behaviour but it is not known whether a similar level of behavioural variation would be observed during on-farm assessments.

## 5. Conclusions

Overall, we identified a high level of intra-observer reliability and a very high level of inter-observer reliability for PC 1, but more varying reliability for PC 2. This study concludes that the QBA approach and the terms included in the *FåreBygg* protocol were reliable for assessing video recordings of sheep behaviour when applied by trained observers, regardless of whether they were a veterinary student, animal welfare inspector or sheep farmer and their previous experience with the methodology. Further work is needed to examine whether similar levels of assessor reliability are achieved when the same terms are applied to assess sheep welfare on-farm, as part of the Fårebygg welfare assessment protocol, and whether the method is sufficiently sensitive to detect differences in the behavioural expression between different housing types, management practices and stockperson handling.

## Figures and Tables

**Figure 1 animals-09-00569-f001:**
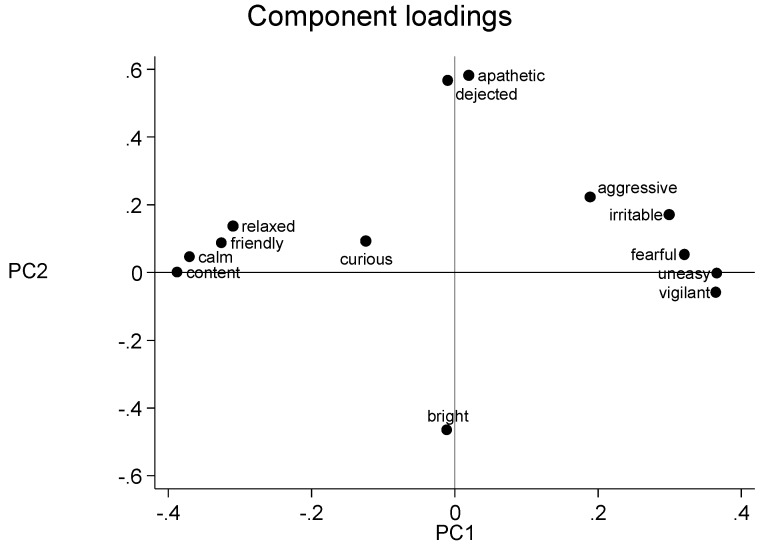
Loading plots generated by the first scoring session for all observers. The x-axis represents principal component 1 and the y-axis represents principal component 2.

**Figure 2 animals-09-00569-f002:**
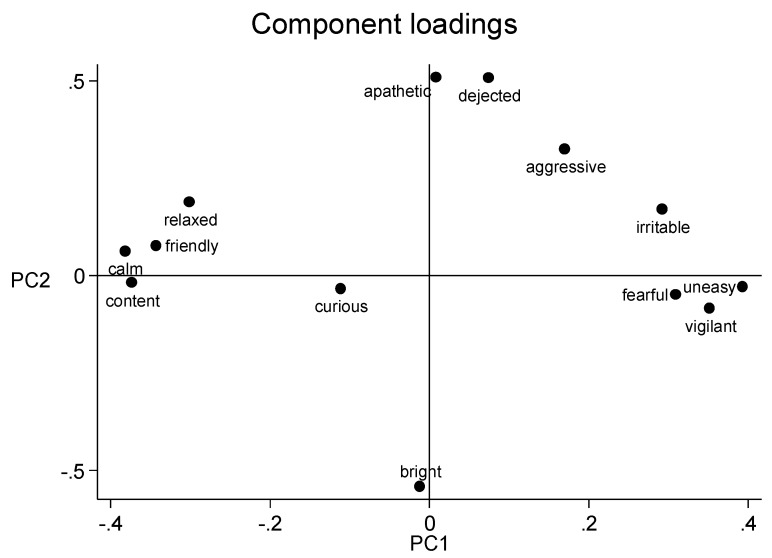
Loading plots generated by the second scoring session for all observers. The x-axis represents principal component 1 and the y-axis represents principal component 2.

**Table 1 animals-09-00569-t001:** The definitions used for the behavioural descriptors in this study.

Behavioural Descriptor	Definition from Protocol
Calm	Not nervous No noticeable reaction to the presence of humans or other animals Does not apply to animals that are sick (lethargic)
Vigilant	Tense, nervous, shy or timid Stops ruminating
Friendly	Sociable Involved in positive social interactions; Social grooming, sniffing other animals, rubbing against one another, lying side by side *and/or* More subtle social interactions, harmonic, positive interest in each other, positive synchronous behaviour Absence of negative social interactions
Fearful	Animals that are showing obvious signs of fear Trying to flee, panting or withdrawing to the far corner/end of the pen
Apathetic	Indifferent, lack of interest in surroundings and/or discouraged Not very responsive The term has a more negative meaning than «dejected»
Aggressive	Actively engaged in physical conflict with other animals
Content	Satisfied and relaxed or engaged in positive activities For example: Playful, social grooming, eating (if there is no competition for feed or displacements) or resting
Irritable	Frustrated, ruthless, grumpy, threatening
Curious	Showing positive interest and anticipation (not vigilant or watchful), and appearing explorative Animals standing still and showing interest have not stopped ruminatingDoes not include the animals that are observing you neutrally
Uneasy	Stressed, apprehensive, restless or impatient
Bright	Awake and alert, showing positive interest in surroundings Animals that are lethargic or have laboured breathing due to gestation will influence this score negatively
Dejected	Animals that are obviously ill or depressed Lack of interest in surroundings Not very responsive Lethargic In pain
Trustful	Showing positive interest in the observer, relaxed, not trying to flee Are not disturbed by the presence of the observer
Relaxed	Resting, either while standing up or lying down Ruminating with a relaxed eye expression (heavy eyelids), not paying much attention to the observer (not curious or vigilant).

**Table 2 animals-09-00569-t002:** Description of the videos used in this study. The order of presentation was different in the two scoring sessions.

Video Number	Video Description	Order of Presentation in Trial	
		1st Session	2nd Session
1	A group of ewes and lambs in an indoor pen. Most of the animals are lying down and ruminating, a few are walking about.	1	5
2	Sheep walking/running in an outdoor pen. A stockperson is moving the animals using a long stick.	2	14
3	A small pen in a sheep house where the ewes are eating and the lambs are in the background.	3	2
4	A small pen in a sheep house where a stockperson is giving concentrate to the ewes in a pen, and lambs are running behind the ewes.	4	17
5	Three adult pregnant ewes in a small indoor pen.	5	4
6	A sheep in a small indoor pen.	6	16
7	The same herd and section in the sheep house as in video 1. A stockperson is distributing hay on the floor.	7	9
8	Ewes and lambs in an indoor pen. Most of them are standing up, a few are lying down.	9	12
9	Same farm and same position as in video 1 and 7. All the ewes are eating concentrate, while the lambs are running around.	10	3
10	Ewes and lambs in an indoor pen. Most are sleeping, some are resting.	11	8
11	Ewes in an indoor pen. Most are lying down and ruminating with their eyes closed.	12	10
12	Ewes in a large indoor pen eating hay. A couple of animals are moving around behind the others, trying to get a place by the feeding trough.	13	6
13	Ewes and lambs in a small indoor pen. The ewes are either lying down or standing still, while some of the lambs are walking/ jumping about.	14	13
14	One adult ewe and two lambs in a field. The ewe is lying down, and the lambs are holding their heads against hers.	15	7
15	Same farm as in videos 1, 7 and 10. The animals are either walking around, eating or interacting with each other.	16	11
16	Adult sheep walking in a shed with straw bedding. They suddenly stop walking and some of them lower their heads quickly.	17	15

**Table 3 animals-09-00569-t003:** Inter-observer agreement given as Kendall’s coefficient of concordance (W) for five groups of observers in two different scoring sessions. The groups were constructed according to background and the level of experience with QBA. PC = principal component.

Session	Group	PC 1		PC 2	
		*W*	*p*	*W*	*p*
1	Veterinary students	0.96	0.0175	0.91	0.0270
	NFSA inspectors	0.91	0.0270	0.55	0.3468
	Veterinary Students and NFSA inspector	0.86	0.0000	0.64	0.0008
	Sheep farmers	0.95	0.0191	0.82	0.0540
	All observers	0.87	0.0000	0.45	0.0004
2	Veterinary students	0.97	0.0155	0.93	0.0229
	NFSA inspectors	0.76	0.0883	0.73	0.1109
	Veterinary Students and NFSA inspector	0.82	0.0000	0.69	0.0003
	Sheep farmers	0.96	0.0170	0.81	0.0600
	All observes	0.85	0.0000	0.65	0.0000

**Table 4 animals-09-00569-t004:** Intra-observer agreement for individual observers given as Kendall’s coefficient of concordance (W). *p*-values are considered significant at 0.05 level. PC = principal component, NFSA = Norwegian Food Safety Authority.

Observer	PC 1		PC 2	
	*W*	*p*	*W*	*p*
Veterinary student 1	0.98	0.0136	0.93	0.0223
Veterinary student 2	0.97	0.0157	0.93	0.0229
NFSA inspector 1	0.91	0.0273	0.67	0.1708
NFSA inspector 2	0.89	0.0306	0.45	0.5603
Sheep farmer 1	0.97	0.0162	0.61	0.2468
Sheep farmer 2	0.92	0.0241	0.58	0.2940

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
