# Peer review of "Intra- and Inter-Observer Reliability of Qualitative Behaviour Assessments of Housed Sheep in Norway"

_animals, 2019, doi:10.3390/ani9080569_

Round 1

Reviewer 1 Report

Animals-548076-peer review

In the present paper the authors test the inter- and intra-observer reliability of a QBA protocol for sheep. Although the research topic in itself is interesting, there are several major issues that have to be addressed.

General:

At the end of the introduction, the authors state that it is important to “assess whether QBA is reliable and is externally valid for application by different stakeholders with different backgrounds in sheep health and welfare…”. I completely agree on this point, especially since QBA is a qualitative method that may be scientifically questionable for several reasons (s. also below). However, I think a sample size of a total of n= 6 observers with different backgrounds and n= 2 observers per group with the same background is far to small to calculate any reliable correlation or to draw any conclusion on inter-observer reliability. Particularly, since the study was conducted by using video files, it is easily feasible and convenient to enlarge the group of observers. It is misleading that the authors talk of “groups” of observers in the discussion when there are actually “pairs” of observers.

In the discussion, the authors mention a frequent criticism of the QBA method, i.e. that it may be very subjective and a mere assignment of anthropomorphic terms to an animal. I think the authors should elaborate more on this, e.g. which descriptors are feasible in terms of a reflection of the “real” emotional state of the animal? Are the most feasible descriptors related to rather negative emotional states (s. also ll. 96-106) which can be verified by additional measurements (e.g. heart rate etc.)? In addition, there may be a further downside of the QBA method that needs to be discussed: while QBA is a qualitative method, the VAS are quantitative. Is it really possible to “translate” a qualitative indicator into a quantitative variable?

The definition of the behavioural descriptors is very vague. E.g. what is meant by “positive activities”? What expressions/features/behavioural signs (e.g. position of the ears, tail, eye movements…) make you conclude that a sheep is “not nervous”? With such vague definitions, the descriptors and thus the QBA remains indeed very subjective and a mere assignment of anthropomorphic terms (a problem the authors mention themselves in ll. 354). This is a pity, as more explicit definitions for the behavioural descriptors could improve the objectivity, the reprocuability, and the overall confirmability of the method.

Line remarks:

Ll. 28-30:             The terms “intra-observer reliability”, “Principle Component Analyses” and “Kendall’s coefficient of concordance” without further explanation are not adequate in a simple summary. Please rephrase in more common language.

Ll. 65-66:             Reference? Is this true for all parts of the world and for all groups within a certain society?

Ll. 175-176:         Which breed was housed? Were the sheep also housed indoors?

Ll. 183-196:         A separate VAS was used for each descriptor. Was there also a separate VAS for each descriptor and individual animal in the videos? I think that it is very likely that not all the animals in a group show the same level of e.g. "alertness", especially not when measured as "mm" on a VAS!

L. 335:             Is there a risk of being “intuitive” when assessing animal welfare states?

Author Response

Dear reviewer,

Thank you very much for your comments and input to our manuscript. You gave us many helpful considerations that will give us a great chance to improve the manuscript. We have made some changes to the text according to your suggestions, highlighted in yellow. Apart from that, answers are given below, in red, marked line to line.

General:

At the end of the introduction, the authors state that it is important to “assess whether QBA is reliable and is externally valid for application by different stakeholders with different backgrounds in sheep health and welfare…”. I completely agree on this point, especially since QBA is a qualitative method that may be scientifically questionable for several reasons (s. also below). However, I think a sample size of a total of n= 6 observers with different backgrounds and n= 2 observers per group with the same background is far to small to calculate any reliable correlation or to draw any conclusion on inter-observer reliability. Particularly, since the study was conducted by using video files, it is easily feasible and convenient to enlarge the group of observers.

To answer this question, we refer to academic editor’s comment about sample size. Several published studies have the same amount of observers, or fewer.

Editor’s comment: In my opinion, this is an interesting and well written paper. I agree with both reviewers that a more detailed description of the descriptors would be useful to strenghten the results and to increase the replicability of the study. Some additional methodological details could be added, as suggested by the reviewers, to improve the manuscript. I do not agree with rev. 1 that 6 observers are not enough to test inter-observer reliability: this is a good sample size, and other papers on inter-observer reliability have already been published based on less than 6 observers. Therefore, in my opinion, this manuscript could be acceptable for publication after minor revision, addressing mainly the methodological issues raised by the reviewers.

It is misleading that the authors talk of “groups” of observers in the discussion when there are actually “pairs” of observers.

Groups and pairs: In this study, we compared the results between different observer pairs and groups. (1) Veterinary students, (2) farmers and (3) NFSA inspectors consisted of pairs.  (4) Vet students + NFSA observers and (5) all observers consisted of more than 2 people and are therefore referred to as groups. We have made some changes in the text to clarify this.

In the discussion, the authors mention a frequent criticism of the QBA method, i.e. that it may be very subjective and a mere assignment of anthropomorphic terms to an animal. I think the authors should elaborate more on this, e.g. which descriptors are feasible in terms of a reflection of the “real” emotional state of the animal? Are the most feasible descriptors related to rather negative emotional states (s. also ll. 96-106) which can be verified by additional measurements (e.g. heart rate etc.)?

The method of QBA used in our study is based on a whole-animal approach and we used a protocol that was developed for another study (FåreBygg). The objective was to evaluate its reliability between and within observers. Behavioral descriptors are not evaluated individually, and to investigate or discuss the validity of each descriptor separately was not the objective of this study. We have made a change in the methods section, where we in a more detailed manner describe the selection of the included behavioral descriptors (L.141-148). The earlier validation of QBA as a whole is described in L.98-106 for different species, and it has been considered sufficiently scientific to be included in e.g. the WQ protocols, and has been used in many scientific studies.

In addition, there may be a further downside of the QBA method that needs to be discussed: while QBA is a qualitative method, the VAS are quantitative. Is it really possible to “translate” a qualitative indicator into a quantitative variable?

This is strictly speaking a question of validity of the method as a whole (where VAS is the standard measurement scale used), and we have provided background information on the validity of QBA in our introduction. The VAS scale helps to translate a qualitative observation into numbers that can be compared by Kendall’s coefficient of concordance.  A VAS scale has no divisions and its use is suggested for things that have an underlying continuum or are hard to quantify, such as pain, where only a minimum and maximum is marked. Both multi-item and single-item VAS has long traditions in other fields of research to measure subjective phenomena in self or others, eg. pain, mental health issues, quality of life etc. The use of VAS scale in this study has the same purpose - any scale with discrete jumps would be more complex, because it would require an exact description for each place of the scale. Furthermore, it has been suggested by several authors that a qualitative method to assess animal welfare is needed to complement quantitative measurements. A VAS scale is according to our judgement the best option for this study –it was chosen exactly because its reliability and our results support the idea that it is possible to reliably “translate” qualitative indicators to numeric results.

The definition of the behavioural descriptors is very vague. E.g. what is meant by “positive activities”? What expressions/features/behavioural signs (e.g. position of the ears, tail, eye movements…) make you conclude that a sheep is “not nervous”? With such vague definitions, the descriptors and thus the QBA remains indeed very subjective and a mere assignment of anthropomorphic terms (a problem the authors mention themselves in ll. 354). This is a pity, as more explicit definitions for the behavioural descriptors could improve the objectivity, the reprocuability, and the overall confirmability of the method.

Another reviewer also made a similar comment about the need to better explain the descriptors. Most studies on QBA have not used detailed definitions of each descriptor – they use only the descriptor. For example, The WQ protocols do not provide a description/definition of the individual behavioural terms, but leaves it up to the observer to interpret the meaning of these. In the first submission, we presented an overview with some keywords. In the resubmission, we will present the full translation of the definitions of the behavioral descriptors. QBA’s validity has been investigated for several species. The point of the method is that one must not use exact parameters to evaluate the animals, or count nor quantify, which is why this is not a part of the description or instruction for the observers in our study. Also, the results of this and other studies suggest that the observers interpret the animals’ behaviour in a very similar way, without thorough details being part of the protocol. 

Line remarks:

Ll. 183-196:         A separate VAS was used for each descriptor. Was there also a separate VAS for each descriptor and individual animal in the videos? I think that it is very likely that not all the animals in a group show the same level of e.g. "alertness", especially not when measured as "mm" on a VAS!

In the protocol used, the animals are evaluated as a group, which is the common approach for QBA in on-farm welfare assessments. Although individual animals act differently, this is the most feasible way to include the QBA in a protocol that investigates animal welfare in large groups of animals. In this study, all of the animals were observed and scored together in all videos, except for one, where three of the animals were observed. This is described in material and methods (L.197-199).  We also explain how the observers were instructed to use the VAS to cover the whole group and the full observation time (L.195-196). We also added a more detailed explanation of the full VAS scale in the resubmission (L.193-195). We hope this clarifies.

335:             Is there a risk of being “intuitive” when assessing animal welfare states?

We believe that you are raising an important issue (subjectivity, antromorphic), which would be of particular concern if the method would be used to assess animal welfare alone, without any additional measurements. However, this is usually not the case, as QBA is used to supplement other, quantitative methods. We believe that there is a greater risk of overlooking important information about the welfare state of animals if no qualitative judgements

Ll. 65-66:             Reference? Is this true for all parts of the world and for all groups within a certain society?

We have made a change in the phrasing according to your concerns, and included some references (L.63-65) 

Ll. 175-176:         Which breed was housed? Were the sheep also housed indoors?

We have addressed this question in a new phrasing (L.179-180). Table 2 presents details about the housing.

Again, thank you for the opportunity to revise this manuscript. We look forward to receiving your comments on the improvements and changes made following first stage of peer review.

Kind regards,

Sofia Diaz-Lundahl (corresponding author, on behalf of all authors)

Reviewer 2 Report

The manuscript is well written, the results are presented in a comprehensible manner. Overall, the paper could be of interest to the target audience of the journal.

Author Response

Dear Reviewer, 

Thank you for taking your time to read and evaluate our manuscript. The comments and suggestions from our reviewers gave us a great chance to improve the manuscript

In the new submission, the changes are highlighted in yellow. 

We look forward to receiving your comments on the improvements and changes made following first stage of peer review.

Kind regards,

Sofia Diaz-Lundahl (corresponding author, on behalf of all authors)

Reviewer 3 Report

This is a very well written paper supported with a wide range of references covering a large range of species. The findings are of use to multiple species.  Well done for producing such a clear article. Please see detailed comments attached.

Author Response

Dear reviewer,

Thank you very much for your input and comments on our manuscript. Your suggestions of changes were very useful, and will help us improve the manuscript. You provided us with some language and paragraph improvements. We have made changes according to many of these. All changes are highlighted in yellow. We also wanted to specify a response for your last comment:

Conclusion, end comment: ? Put into the context of resource- or animal- based indicators. And

404 potential use when considering welfare with respect to the Five Domains Model?

Thank you for highlighting how we could improve the text. We have used your comments to amend the conclusion (Lines 409-413), which states that:

Further work is needed to examine whether similar levels of assessor reliability are achieved when the same terms are applied to assess sheep welfare on-farm, as part of the FåreBygg welfare assessment protocol, and whether the method is sufficiently sensitive to detect differences in the behavioural expression between different housing types, management practices and stockperson handling.

We believe that further evaluation of the mix between animal-based and resource-based indicators belongs in conclusions of the main project FåreBygg for which the protocol was developed. I will pass the suggestion on to the authors of this article.

Again, thank you for the opportunity to revise this manuscript. We look forward to receiving your comments on the improvements and changes made following first stage of peer review.

Kind regards,

Sofia Diaz-Lundahl (corresponding author, on behalf of all authors)

Round 2

Reviewer 1 Report

Dear authors,

I am happy to see that you have agreed to my comments from the first revision round. I think that the manuscript has been significantly improved. Particularly by defining the descriptors in more detail, e.g. to clarify that "calm" definitely not applies to "sick animals (lethargic)". It is now also much clearer when "pairs" and "groups" of observers are compared.